# Intrinsic Disorder and Phosphorylation in BRCA2 Facilitate Tight Regulation of Multiple Conserved Binding Events

**DOI:** 10.3390/biom11071060

**Published:** 2021-07-20

**Authors:** Manon Julien, Rania Ghouil, Ambre Petitalot, Sandrine M. Caputo, Aura Carreira, Sophie Zinn-Justin

**Affiliations:** 1Institute for Integrative Biology of the Cell (I2BC), CEA, CNRS, Université Paris-Sud, 91190 Gif-sur-Yvette, France; Manon.julien@i2bc.paris-saclay.fr (M.J.); rania.ghouil@i2bc.paris-saclay.fr (R.G.); 2L’Institut de Biologie Intégrative de la Cellule (I2BC), UMR 9198, Paris-Saclay University, 91190 Gif-sur-Yvette, France; aura.carreira@curie.fr; 3Service de Génétique, Unité de Génétique Constitutionnelle, Institut Curie, 75005 Paris, France; ambre.petitalot@curie.fr (A.P.); sandrine.caputo@curie.fr (S.M.C.); 4Institut Curie, Paris Sciences Lettres Research University, 75005 Paris, France; 5Unité Intégrité du Génome, ARN et Cancer, Institut Curie, CNRS UMR3348, 91405 Orsay, France

**Keywords:** disorder, phosphorylation, cancer, DNA repair, mitosis, meiosis, variants, NMR, protein-protein interaction, 3D structure

## Abstract

The maintenance of genome integrity in the cell is an essential process for the accurate transmission of the genetic material. BRCA2 participates in this process at several levels, including DNA repair by homologous recombination, protection of stalled replication forks, and cell division. These activities are regulated and coordinated via cell-cycle dependent modifications. Pathogenic variants in *BRCA2* cause genome instability and are associated with breast and/or ovarian cancers. BRCA2 is a very large protein of 3418 amino acids. Most well-characterized variants causing a strong predisposition to cancer are mutated in the *C*-terminal 700 residues DNA binding domain of BRCA2. The rest of the BRCA2 protein is predicted to be disordered. Interactions involving intrinsically disordered regions (IDRs) remain difficult to identify both using bioinformatics tools and performing experimental assays. However, the lack of well-structured binding sites provides unique functional opportunities for BRCA2 to bind to a large set of partners in a tightly regulated manner. We here summarize the predictive and experimental arguments that support the presence of disorder in BRCA2. We describe how BRCA2 IDRs mediate self-assembly and binding to partners during DNA double-strand break repair, mitosis, and meiosis. We highlight how phosphorylation by DNA repair and cell-cycle kinases regulate these interactions. We finally discuss the impact of cancer-associated variants on the function of BRCA2 IDRs and more generally on genome stability and cancer risk.

## 1. Introduction

The BReast CAncer protein 2 (BRCA2) is a ubiquitous protein essential for embryonic development: BRCA2 knockout mice show early embryonic lethality and hypersensitivity to irradiation [1]. Depletion of BRCA2 also causes a wide range of defects in DNA repair and recombination [2], protection of stalled replication forks [3], regulation of telomere length [4,5], mitosis [6,7], meiotic recombination, and fertility [8]. At the molecular level, BRCA2 interacts with the strand exchange (or recombinase) protein RAD51 (for RADiation sensitive protein 51) and facilitates its function in various Homologous Recombination (HR) contexts [1,9,10,11]. It is essential for RAD51-mediated HR in both mitotic and meiotic cells. A defect in BRCA2 HR function causes genetic instability and increases cancer risk. In addition, BRCA2 has a protective function during replicative fork stalling that is mechanistically distinct from repair by HR [3]. It prevents degradation of nascent strands at stalled forks by stabilizing RAD51 filaments. BRCA2 variants with compromised fork protection exhibit increased chromosomal aberrations. Finally, during mitosis, BRCA2 inactivation impairs the completion of cell division, thus triggering alterations in chromosome number and abnormalities such as centrosome amplification [6,12]. Biallelic loss-of-function variants in *BRCA2* cause Fanconi’s anemia type D1, which is a rare autosomal recessive cancer susceptibility disorder characterized by an increased number of chromosomal breaks after cells are exposed to DNA-damaging agents [13,14]. Pathogenic variants in one *BRCA2* allele confer higher risk of breast and/or ovarian cancers [15,16,17], as well as pancreatic and prostate cancers [18].

Despite the accumulated knowledge about BRCA2 essential functions, the molecular mechanisms associated with these functions are poorly described. This is in part due to the disordered character of BRCA2, which is examined in this review. Within the 3418 amino acids (aa) of BRCA2, one single domain of 700 residues is folded, as revealed by the analysis of the crystal structure of the mouse and rat homologous domains bound to a single-stranded DNA (ssDNA) and in interaction with the short acidic protein DSS1 (for Deleted in Split hand/Split foot protein 1) [19] (Figure 1A,B). Recently, several groups purified full-length BRCA2 and performed negative-staining electron microscopy, scanning force microscopy and fluorescence based single molecule analyses of this protein either free or bound to RAD51, DSS1, and ssDNA.

A large range of particle sizes was observed [21,22,23]. Selection of a subset of these particles led to the generation of 3D reconstructions for a dimeric form of full-length BRCA2 free and in complex with RAD51 [21]. Free BRCA2 was described as an elliptical dimeric molecule of 25 nm × 13 nm × 12 nm, with its folded domains localized at both vertexes of the ellipse [21]. In the presence of RAD51, the dimeric BRCA2 molecules formed a larger ellipse of 26 nm × 16 nm × 16 nm, with extra density present next to the region identified as corresponding to the BRC repeats. However, how the BRCA2 dimers assemble into these ellipses, the folded domains from each monomer being connected through structurally less characterized regions of BRCA2, is still unclear. Further electron microscopy, scanning force microscopy, and quantitative single-molecule fluorescence studies showed that purified full-length BRCA2 forms heterogeneous oligomeric structures, assembled at least in part through its *N*-terminal region BRCA2_1–714_ [24] or its central region around Phe1524 [23]; they revealed that these structures exhibit remarkable rearrangement by RAD51, DSS1, and ssDNA [22,23,24]. The BRCA2 apparent structural plasticity is a hallmark of proteins with intrinsically disordered regions (IDRs).

Several studies have reported that cancer-associated proteins are rich in IDRs [25,26]. These protein regions lack a stable secondary structure. They interact with their partners through interfaces that never reach the size of the largest interfaces of ordered complexes but are characterized by a specifically large surface per residue: IDRs use a larger portion of their surface for interaction with their partner, sometimes 50% of the whole, as opposed to only 5–15% for most ordered proteins [27,28]. In addition, because the free energy arising from the contacts between an IDR and its target is reduced by the free energy needed to fold the IDR, interactions with IDRs enable high specificity coupled with moderate affinities [28,29]. Finally, IDRs are highly accessible to enzymes, and thus rich in post-translational modifications (PTMs). Such PTMs represent transient events that regulate interactions mediated by IDRs. Functionally relevant binding sites as well as PTMs are generally conserved in IDRs, which facilitates their identification using bioinformatics tools [30,31]. Here, we review the predicted as well as experimentally demonstrated structural and functional properties of BRCA2 IDRs, with a special focus on phosphorylation sites regulating BRCA2 binding to partners. We also discuss the impact of BRCA2 variants identified in patients with breast and/or ovarian cancers, which are detected throughout the whole BRCA2 protein, including the IDRs. Most of these variants are of uncertain clinical significance (VUS), especially when they are detected in few patients and are located in poorly characterized IDRs. 

## 2. BRCA2 Exhibits a Set of Conserved Motifs Predicted to Be Disordered

Analysis of BRCA2 protein sequence using disorder prediction tools provided a general view of the structural organization of this protein. We predicted BRCA2’s propensity for disorder using SPOT-Disorder2 [20], the best program for discriminating between order and disorder as described in Nielsen et al. [32]. The resulting plot unambiguously showed that the ssDNA- and DSS1-binding domain is the unique folded domain of BRCA2, here named DBD for DNA Binding Domain (Figure 1C). Short regions with an intermediate disorder propensity were also identified. In the central region of BRCA2, these regions correspond to the eight BRC repeats (so called because it is repeated in BRCA2), which bind to the recombinase RAD51 and its meiotic-specific paralog DMC1 [11,33]. The fourth BRC repeat, BRC4, was crystallized when bound to the ATPase domain of RAD51: it remains in contact with RAD51 over a stretch of 28 aa (residues Leu1521 to Glu1548), and forms a β-hairpin and an α-helix in the complex, as shown in Figure 1D [11]. In order to further identify functional domains in the predicted disordered BRCA2 regions, we aligned the sequences of 24 BRCA2 homologs, from human to fishes. We used Jalview 2.10.1 to calculate a conservation score per residue reflecting the conservation of the amino acid physico-chemical properties at each position in the BRCA2 sequence [34]. Figure 1E shows that the largest conserved region corresponds to the *C*-terminal DBD domain. However, small patches of conserved motifs are also revealed in the regions that are predicted to be disordered. In particular, BRC1, BRC2, BRC4, BRC7, and BRC8 are highly conserved, whereas the three other BRC repeats are only moderately conserved. Finally, such analysis demonstrated that several other motifs are predicted to be disordered and conserved. One of these motifs was already functionally characterized: the *N*-terminal region between Leu24 and Ser37 forms a short α-helix interacting with a hydrophobic pocket at the surface of the partner and localizer of BRCA2 (PALB2) (Figure 1F) [35]. This interaction is essential for BRCA2 localization at double-strand break (DSB) repair foci during HR. Other motifs have only been recently characterized. In the following sections, we will first focus on the functional role of the well-studied disordered motifs binding to RAD51 and PALB2, and then we will concentrate on a few of the new motifs identified more recently by different groups including us.

## 3. Structural Studies Illustrate the Role of BRCA2 Disordered Regions in DNA Repair by HR

BRCA2 contributes to genome integrity by being an essential factor of the HR pathway [36]. HR is required for the repair of DSBs, inter-strand crosslinks, and replicative DNA lesions using the genetic information from the sister chromatid as a template. Thus, HR takes place during S and G2 phases. Defects in this process generate unrepaired DNA DSBs, which are highly toxic for the cell, leading to genome instability, and increased risk of cancer.

The recruitment of BRCA2 to DNA damage sites is mediated by the PALB2 protein, which is itself recruited by BRCA1, presumably at the ssDNA/dsDNA junction produced after resection of the DNA break ends [37,38]. The BRCA2 region involved in PALB2 binding includes residues 24 to 37 [35]. As illustrated in the section above, the *N*-terminal region of BRCA2 is predicted to be disordered in solution. However, the crystal structure of the complex between the BRCA2 peptide from Leu24 to Ser37 and PALB2 showed that BRCA2 folds upon binding (Figure 1F; [35]). The localization of BRCA2 via PALB2 is essential for efficient HR. PALB2 mutants with impaired BRCA2 binding decrease the capacity for DSB-repair by HR and increase cellular sensitivity to ionizing radiation [39].

After its recruitment to the DNA damage locus, BRCA2 facilitates the displacement of the ssDNA binding protein Replication Protein A (RPA) and the loading of the recombinase RAD51 onto ssDNA. RPA binding is a pre-requisite for RAD51 filament formation, as it contributes to remove DNA secondary structures and protect DNA from nucleolytic degradation; however, RPA’s high affinity for ssDNA is also an impediment for RAD51 nucleation [40,41]. The folded domain of BRCA2 binds to the acidic protein DSS1 (Figure 1B), which interacts with the DNA binding surface of RPA and favors RPA’s release from ssDNA [19,42]. Moreover, BRCA2 central region, which is predicted to be disordered, contains eight conserved repeats, approximately 35 aa in size, the BRC repeats, which have the capacity to bind RAD51 (Figure 1A). As illustrated for BRC4 (from Leu1521 to Glu1548), the BRC motif folds upon interaction with the RAD51 ATPase domain (Figure 1D; [11]). BRC4 blocks ATP hydrolysis by RAD51; it favors nucleation of the ATP-bound form of RAD51 onto ssDNA, preventing its assembly on dsDNA [43]. The BRC repeats display different affinities for RAD51. Only the first four repeats have high affinity for monomeric RAD51 [44]. The other four repeats bind with low affinity to monomeric RAD51 and high affinity to the RAD51-ssDNA nucleoprotein filament [44]. Through these molecular interactions, BRCA2 favors the formation of a stable right-handed helical nucleo-protein filament of RAD51 on ssDNA, which is the active form for the search of sequence homology in the intact copy of the chromosome. Upon encountering a homologous sequence, the nucleo-protein filament pairs with the complementary strand, resulting in the displacement of the non-complementary strand from the duplex to generate a D-loop structure (displacement loop) and promote DNA strand exchange.

In addition to its role in somatic cells, BRCA2 also contributes to HR during meiosis. The core HR machinery (BRCA2, PALB2, RAD51) is extended in meiosis with a set of meiosis-specific proteins, such as the ssDNA-binding proteins MEIOB and SPATA22 and the recombinase DMC1. BRCA2 interacts with both RAD51 and the meiosis-specific recombinase DMC1. It binds DMC1 via the RAD51-binding BRC repeats [33,45] and a DMC1-specific site located in a region predicted to be disordered between Ser2386 and Lys2411 [46]. However, no biophysical characterization of the interaction between BRCA2 and meiosis-specific proteins has been reported until recently.

## 4. During Interphase, the *C*-Terminal Disordered Region of BRCA2 Is Required for the Protection of Stalled Forks

BRCA2 is also essential for the protection of stalled replication forks (RF). Following replication stress, RF are slowed or stalled and, if not protected, newly replicated DNA can undergo unscheduled degradation by nucleases such as MRE11 [47]. BRCA2 has a protective function during replication fork stalling that is mechanistically distinct from repair by HR [3]. The C-terminal disordered and conserved region of BRCA2, from Ala3270 to Gly3305, binds to and stabilizes RAD51-ssDNA filaments, thus preventing nascent-strand degradation [3,48,49]. BRCA2 mutants with compromised fork protection exhibit increased spontaneous and HU-induced chromosomal aberrations that are alleviated by MRE11 inhibition.

Ser3291, in the BRCA2 C-terminal region binding to RAD51-ssDNA filaments, is strictly conserved (Figure 1E). It undergoes CDK (for Cyclin Dependent Kinase)-dependent phosphorylation at the G_2_-M phase transition and dephosphorylation upon DNA damage [50]. Phosphorylation of Ser3291 blocks interaction between BRCA2 and oligomeric RAD51. This was demonstrated using two separation of function mutants, S3291A and S3291E, that abrogate RAD51 binding; cells bearing these mutations are defective in replication fork protection but are able to repair DSB by HR. Furthermore, BRCA2 and PALB2 facilitate the recruitment of Polη by directly interacting with the polymerase during the initiation of DNA synthesis [51]. The BRCA2 region from Leu1409 to Asn1596 (including BRC3 and BRC4) is responsible for binding to the polymerase. The trimeric complex stimulates the initiation of recombination-associated DNA synthesis by Polη. Altogether, at blocked replication forks, BRCA2 disordered regions, including its C-terminus, are essential for binding to RAD51 filaments and stimulating the initiation of DNA synthesis by Polη in vitro. Phosphorylation of Ser3291 promotes RAD51 filament disassembly, which in turn promotes entry into mitosis [52].

## 5. Nuclear Magnetic Resonance Analyses Support the Presence of Additional Disordered and Conserved Regions in BRCA2

Within the BRCA2 regions predicted to be disordered, four additional conserved fragments have been recently experimentally characterized: BRCA2_48–284_ [53], BRCA2_250–500_ (unpublished data), BRCA2_1093–1158_ (unpublished data) and BRCA2_2213–2342_ [54] (Figure 2A). These BRCA2 fragments were recombinantly produced in bacteria as ^15^N, ^13^C labeled proteins, purified by chromatography and further analyzed using solution-state Nuclear Magnetic Resonance (NMR), a biophysical technique uniquely suited for the structural characterization of soluble disordered proteins at the residue level [55].

Indeed, whereas X-ray crystallography and cryo-electron microscopy cannot describe the conformations of highly flexible proteins or protein regions retaining a large mobility, NMR provides information on their structural propensities in solution. The very common 2D NMR ^1^H-^15^N Heteronuclear Single Quantum Coherence (HSQC) experiment is used to characterize the local chemical environment of each backbone amide bond atom and thus visualize the disorder propensity of a protein at the residue level. Here, the ^1^H-^15^N HSQC spectra of the four BRCA2 fragments are presented. The narrow dispersion of their ^1^H-^15^N correlation peaks in the ^1^H dimension demonstrated that they are disordered protein regions (Figure 2B). Further analysis of the ^1^H, ^15^N and ^13^C chemical shifts of BRCA2_48–284_, BRCA2_1093–1158_, and BRCA2_2213–2342_ was performed to obtain the secondary structure propensity of these fragments at the residue level. Whereas BRCA2_48–284_, BRCA2_1093–1158_ and BRCA2_2213–2342_ do not form any stable α−helix or β-strand, they contain small motifs adopting transient α-helical structures (Figure 2C).

NMR can also be used to verify that a mutation does not affect the conformation of the fragment of interest, or identify structural defects in a protein mutant associated with a disease. Here, fragment BRCA2_48–284_ contains five cysteines. In IDRs, cysteines are solvent-exposed, and, under our working conditions, are quickly oxidized, thus favoring protein aggregation. We observed that, even in the presence of high concentrations (about 10 mM) of thiol reducer, intra and inter-cysteine bonds are already formed a few hours after the addition of the reducer. Mutation of cysteines into alanine is a common strategy to prevent IDR aggregation in vitro. We recorded 2D NMR ^1^H-^15^N HSQC experiments on the BRCA2_48–284_ fragment either wild-type or with four cysteines mutated into alanine. The two ^1^H-^15^N HSQC spectra are similar, demonstrating that the cysteine mutations do not modify the structural conformation of BRCA2_48–284_; a construct with only one cysteine was further used for phosphorylation and binding studies [53]. Similarly, such strategy can be used to measure the impact of a cancer-associated mutation on the solution structure of a BRCA2 IDR [57].

## 6. BRCA2_48–284_ Phosphorylation by CDKs and PLK1 Ensures Correct DNA Repair before Mitosis and BRCA2 Midbody Localization during Cytokinesis

The characterization of the function of BRCA2_48–284_ started almost 20 years ago [58]. BRCA2_48–284_ contains two conserved patches between residues 63 to 93 and 164 to 230, phosphorylated by CDKs [7,59] and PLK1 (for Polo-Like Kinase 1) [57], respectively (Figure 3A). These phosphorylation events take place at the interphase to mitosis transition and were progressively associated with various mitotic functions of BRCA2.

In the conserved region BRCA2_63–93_, phosphorylation by CDKs at Thr77 was identified by Western blot using an antibody raised against pThr77 [59]. This phosphorylation site belongs to a conserved motif that matches the optimal PLK1 binding motif Ser-[pSer/pThr]-[Pro/X] (Figure 3B). Consistently, a direct in vitro interaction was reported between BRCA2_pThr77_ and PLK1 [59]. This interaction brings together PLK1 and another BRCA2 partner: the RAD51 recombinase. It was proposed that, during the interphase to mitosis transition, BRCA2 acts as a molecular platform that facilitates PLK1-mediated RAD51 phosphorylation at position Ser14 [59]. This last phosphorylation event peaks in mitosis, and enhances the association of RAD51 with stressed replication forks for protecting the genomic integrity of proliferating human cells [59]. It stimulates the subsequent phosphorylation of RAD51 Thr13 by Casein Kinase 2 [59,60]. Di-phosphorylation of RAD51 at Ser14 and Thr13 is also detected in response to ionizing radiation [59]. Phosphorylation at Thr13 triggers direct binding to the FHA domain of NBS1, a component of the MRN complex involved in DSB DNA repair [60]. This interaction helps to increase the RAD51 concentration at the site of damage, and promotes efficient DNA repair by HR before mitosis onset [60]. Finally, other conserved CDK phosphosites characterized by the motif [pSer/pThr]-Pro are detected in BRCA2_63–93_ (Figure 2B). Consistently, we have observed in vitro that CDKs are able to phosphorylate not only Thr77 but also other conserved residues such as Thr64 and Ser93. A deeper characterization of the phosphorylation events taking place in this region may highlight complementary BRCA2-dependent mechanisms for entry into mitosis.

In 2003, Lin et al. identified another BRCA2 region between residues 193 and 207 that is highly phosphorylated in mitosis (Figure 3C; [58]). This region is a target for the kinase PLK1, and only Ser193 was clearly identified as a target of PLK1 in mitosis. However, mutation S193A did not completely abrogate the PLK1-dependent phosphorylation of BRCA1_1–284_, whereas deletion of residues 193 to 207 did abolish phosphorylation, suggesting that other sites are phosphorylated next to Ser193. This highlights a very common situation in the IDP field; IDRs are enriched in phosphorylation sites, creating clusters with multiple sites. Identification of modification sites at very close positions in the protein sequence is challenging for mass spectrometry and Western-blot analyses. Recently, we monitored phosphorylation of BRCA2_167–260_ by PLK1 using 2D NMR ^1^H-^15^N HSQC experiments (Figure 3D; [57]). Upon phosphorylation, the chemical environment of the backbone H, N amide bond atoms of the phosphoresidue is modified, which changes the position of the corresponding ^1^H-^15^N correlation peak in the spectrum (Figure 3D). Thus, four phosphosites were identified: pSer193, pThr207, pThr219, and pThr226 [57]. A function was proposed for the phosphorylation of both strictly conserved residues Ser193 and Thr207 (Figure 3C). The biological role of the phosphorylation of the less conserved Thr219 and Thr226 in human cells still remains to be clarified.

BRCA2 pSer193 was the first phosphosite identified in this BRCA2 region. It was reported as essential for BRCA2 localization at the Flemming body during cytokinesis [7]. Indeed, BRCA2 S193A failed to localize at the midbody, while the phosphomimetic S193E was sufficient to restore BRCA2 localization. Moreover, phosphorylation of Thr77 that triggers PLK1 interaction with BRCA2 was proposed to increase phosphorylation of Ser193 by PLK1 and BRCA2 localization at the midbody. The protein able to recruit BRCA2 phosphorylated on Ser193 at the midbody is currently unknown. The actin-binding protein Filamin A could contribute to this recruitment; however, it interacts with BRCA2 DBD [61]. More work would be required to identify the BRCA2 partner binding to the highly conserved region centered on pSer193 and the strictly conserved hydrophobic residue Trp194 (Figure 3D). 

## 7. BRCA2_199–210_ and BRCA2_1093–1158_ Mediate the Assembly of a Large Complex, Including PLK1, BUBR1, and PP2A, Regulating Chromosome Alignment during Mitosis

We recently confirmed that Thr207 was phosphorylated in mitosis, using an antibody raised against a BRCA2 peptide centered on pThr207 [57]. We also noticed that phosphorylation of BRCA2 Thr207 creates a docking site for the Polo-Box domain (PBD) of PLK1 (PLK1_PBD_). This phospho-dependent interaction was confirmed by Isothermal Titration Calorimetry (ITC): the K_d_ measured between BRCA2_194–210(pThr207)_ and PLK1_PBD_ is close to 0.1 nM, (Figure 3E; [57]), as also reported for other interactions between phosphorylated peptides and PLK1_PBD_ [62], whereas the non-phosphorylated BRCA2 peptide BRCA2_194–210_ does not bind to PLK1_PBD_. The complex between BRCA2_194–210(pThr207)_ and PLK1_PBD_ was crystallized, revealing that, as also described for other phosphorylated peptides, BRCA2_199–210(pThr207)_ binds in the cleft formed by the two Polo boxes of PLK1_PBD_ (Figure 3F). Surprisingly, mutating Thr207 into the phosphomimetic glutamate or aspartate did not generate any interaction with PLK1_PBD_, showing that these mutations cannot be used as phosphomimetics to test the function of Thr207 phosphorylation in cells [57]. In mitosis, binding of BRCA2 pThr207 to PLK1 triggers the assembly of a larger complex containing BRCA2, PLK1, the mitotic checkpoint kinase BUBR1 (for Budding Uninhibited by Benzimidazole-Related 1), and the phosphatase PP2A (for Protein Phosphatase 2A) at the kinetochore (Figure 4A; [57]). Thus, BRCA2 serves as a platform that brings together BUBR1 and PLK1. This favors BUBR1 phosphorylation by PLK1 at tension-sensitive sites, involved in the regulation of the kinetochore-microtubule attachment [63]. In addition, breast cancer variants S206C and T207A impair PLK1 binding and result in unstable microtubule-to-kinetochore attachments, misaligned chromosomes, faulty chromosome segregation and aneuploidy [57]. This highlights the role of BRCA2 in preserving genome integrity during mitosis and shows how a simple phosphorylation event can be essential for genome integrity.

BRCA2 phosphorylation at Thr207 further triggers the assembly of a large complex including PLK1, BUBR1, and PP2A through poorly characterized interactions. Phosphorylation of BUBR1 by PLK1 promotes the interaction of its Leu-X-X-Ile-X-Glu motif with the B56 subunit of PP2A (Figure 4A; [64,65]). BRCA2_1093–1158_, located in between the BRC1 and BRC2 motifs, also exhibits a conserved Leu-X-X-Ile-X-Glu motif that interacts with PP2A (Figure 4A,B; [66]). Thus, BUBR1 and BRCA2 might compete for binding to the same B56 subunit of PP2A. Through these interactions, PP2A protects the kinetochore-microtubule interaction from excessive destabilization by Aurora B [67].

**Figure 4 biomolecules-11-01060-f004:**
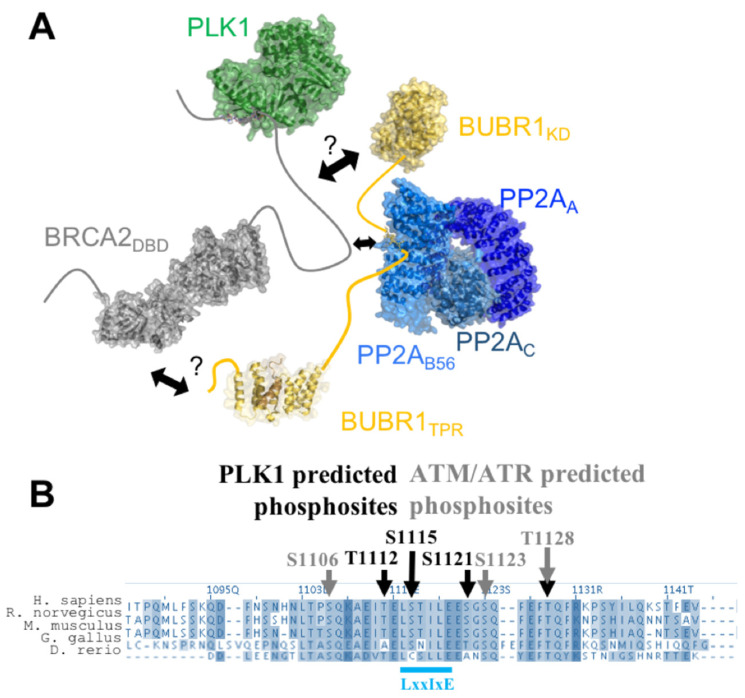
BRCA2_1093–1158_ contains predicted PLK1 and ATM/ATR phosphorylation sites and a PP2-B56 binding site. (**A**) Scheme of the BRCA2/PLK1/BUBR1/PP2A complex, assembled upon phosphorylation of Thr207 by PLK1. Human BRCA2 (in grey) has a long *N*-terminal region, which interacts with PLK1_PBD_ (PDB code: 6GY2) upon phosphorylation of Thr207 by PLK1. No human full-length PLK1 structure has been elucidated yet: here the PLK1 model (green) is built using the *Danio rerio* PLK1 structure (PDB code: 4J7B). BRCA2 interacts with BUBR1 (yellow) through a controversial interface (arrows and question marks): Futamura et al. [68] reported an interaction between BRCA2_2861–3176_ and the kinase domain of BUBR1 (model built from the fly structure; PDB code: 6JKK), whereas Choi et al. [69] reported an interaction between BRCA2_3189–3418_ and the *N*-terminal region of BUBR1 including its TPR domain (residues 57 to 220; PDB code: 3SI5). BRCA2 and BUBR1 (PDB codes: 5JJA; 5K6S; 5SWF) interact with the B56 subunit of the phosphatase PP2A (3D structure of the whole protein formed by three subunits, in blue; PDB code: 2NPP) [64,66,70]. (**B**) Alignment of the human, mouse, rat, chicken and zebra fish sequences homologous to human BRCA2_1093–1158_, highlighting the high conservation of the predicted PLK1 and ATM/ATR phosphorylation sites, as well as the PP2A binding site. Black and grey arrows point to the predicted phosphorylation sites for PLK1 and ATM/ATR, respectively. The docking site for the B56 subunit of the PP2A phosphatase identified in this region [66,71] is mediated by the L-X-X-I-X-E consensus motif (cyan), with X for any amino acid [66].

The conserved region BRCA2_1093–1158_ contains several highly conserved phosphorylation sites, predicted to be targets of either PLK1 or ATM (for Ataxia Telangiectasia Mutated) and ATR (for ATM and RAD3-related) (Figure 4B). A larger BRCA2 region, still located between BRC1 and BRC2, was reported to be phosphorylated by PLK1 during mitosis [72]. As phosphorylation of the Leu-X-X-Ile-X-Glu motif increases binding to the B56 subunit of PP2A [65], phosphorylation by PLK1 could regulate BRCA2 binding to this phosphatase at the kinetochore. Phosphorylation of BRCA2_1093-1158_ by ATM/ATR was also recently experimentally confirmed [71]. The authors showed that phosphorylation of the [Ser/Thr]-Gln motifs, specifically at Ser1123 and Thr1128, increases binding of BRCA2_1113–1129_ to the B56 subunit of PP2A. In addition, mutation of the three [Ser/Thr]-Gln motifs located in this BRCA2 region increases the sensitivity of cells to Poly(ADP-ribose) Polymerase inhibitors and leads to a loss of efficient RAD51 loading and HR-mediated DNA repair. These results revealed another role for the BRCA2/PP2A interaction in DNA repair, and suggested that PP2A-B56 might be a general regulator of BRCA2 function throughout the cell cycle.

## 8. Both Folded and Disordered Regions of BRCA2 Contribute to Its DNA Binding Properties

Purified full-length BRCA2 binds to DNA above a protein concentration of 2 to 20 nM [73,74,75]. It binds to both ssDNA and dsDNA; however, it displays a preference for ssDNA (as well as tailed) substrates over dsDNA [74]. The mouse BRCA2 DBD in complex with the acidic protein DSS1 interacts with ssDNA [19]. Native gel electrophoretic mobility-shift assays showed that it binds to oligo(dT), oligo(dC), and mixed ssDNA sequences but not to dsDNA. The crystal structures of the mouse DBD bound to DSS1 and an oligo(dT)9 illustrates how the oligonucleotide/oligosaccharide-binding 2 and 3 motifs of this domain interact with small oligonucleotides (8 to 12 nt), as observed at concentrations above 3 uM and independently of its tower fragment. The BRCA2 folded domain in complex with DSS1 further binds with a 10-fold higher affinity to larger oligonucleotides (32 to 36 nt) in the presence of its tower fragment [19]. The disordered BRCA2 region from residue 250 to 500 also contributes to DNA binding [75]. It is less conserved than the other disordered regions described in this review (Figure 1 and Figure 5A).

A Maltose-Binding Protein tagged BRCA2 WT fragment MBP-BRCA2_250–500_ produced in HEK293 human cells binds to both ssDNA and dsDNA at a concentration around 1 μM, depending on the oligonucleotide sequence; it is the only BRCA2 fragment yet demonstrated to bind dsDNA [75]. When produced in bacteria as a variant in which all cysteines are mutated into alanine or serine (for solubility purposes, see above), BRCA2_250–500_ is disordered in solution (Figure 2B and Figure 5B). Cysteines were shown as important for DNA binding to various DNA forms [75]. Consistently, the variant BRCA2_250–500_ produced in bacteria binds to ssDNA and dsDNA at higher (micromolar) concentrations (Figure 5C,D). It shows a preference for long ssDNA versus short ssDNA, and also binds to short dsDNA in our experiments (Figure 5C,D). This provides evidence that cysteines contribute to DNA binding but are not mandatory.

In cells, MBP-BRCA2_250–500_ significantly enhances RAD51 recombination activity: as the full-length BRCA2 protein, it stimulates RAD51-driven DNA strand exchange reaction in the presence (and only in the presence) of RPA. Full-length BRCA2 increases RAD51-mediated strand exchange at a 10-fold lower concentration, whereas the DBD shows only a weak stimulating effect, observable at a 30-fold higher concentration [19,75]. In the case of the DBD, this weak effect depends on its capacity to present DSS1 that exposes a negatively charged fragment competing with ssDNA for binding RPA [42]. In the case of BRCA2_250–500_, which exhibits an isoelectric point of 5.3, both its reported capacity to bind DNA, potentially through its positively charged regions, and its still hypothetical capacity to mimic DNA, because of its negatively charged patches, could contribute to displace RPA from DNA.

## 9. A Conserved and Disordered BRCA2 Region Contains a Cryptic Repeat That Recruits HSF2BP, Thus Triggering BRCA2 Degradation

A recent search for new BRCA2 partners identified HSF2BP (also named MEILB2) as another BRCA2-binding protein, endogenously expressed in meiotic cells, and ectopically produced in cancer cells [76,77]. HSF2BP is required for meiotic HR during spermatogenesis; its disruption abolishes the localization of RAD51 and DMC1 recombinases in spermatocytes, leading to errors in DSB repair by meiotic HR and consequently male sterility [78]. HSF2BP exhibits an *N*-terminal α-helical oligomerization domain and a C-terminal Armadillo domain [76,79]. Within BRCA2_2213–2342_, a highly conserved IDR binds to the Armadillo domain of HSF2BP (HSF2BP_ARM_) (Figure 6A; [54,76]). NMR analysis confirmed that BRCA2_2213–2342_ is disordered in solution (Figure 2B) and that it directly interacts with HSF2BP_ARM_: addition of unlabeled HSF2BP_ARM_ to an ^15^N-labeled BRCA2_2213–2342_ caused a global decrease of the intensities of the NMR 2D ^1^H-^15^N HSQC peaks, with region from aa 2252 to aa 2342 showing the largest decrease (Figure 6B; [54]). Unexpectedly, ITC experiments revealed that this interaction is characterized by a high affinity, on the nanomolar range, which is the largest affinity ever measured for a BRCA2 interaction [54].

In addition, unexpectedly, our crystal structure of the complex showed that a dimeric HSF2BP_ARM_ (here chains A, D and B, C) further dimerizes through two BRCA2 peptides in order to form a tetramer (Figure 6C). The BRCA2 peptides (in magenta) run along the V shape groove formed by either chains A and C or chains B and D. Each peptide interacts through its *N*-terminal region with one HSF2BP_ARM_ monomer and through its C-terminal region with another HSF2BP_ARM_ monomer. These BRCA2 peptide regions are encoded by exons 12 and 13, respectively, but they contain the same sequence motif critical for binding to the conserved groove of an HSF2BP_ARM_ monomer (Figure 6C,D; [54]). Such repeated motif is responsible for HSF2BP_ARM_ tetramerization upon BRCA2 binding. Deletion of the motif encoded by exon 12 impaired tetramerization and caused a 1000-fold loss in affinity, leading to a micromolar affinity between HSF2BP and BRCA2. However, it did not impair meiotic HR and caused no fertility defect in mice [54]. Thus, the role of the high affinity interaction between HSF2BP and BRCA2 in meiosis is still unclear. In somatic cells, this interaction interferes with the role of BRCA2 in DNA inter-strand crosslink repair: it triggers proteasome-dependent degradation of BRCA2 [77]. More generally, the high affinity interaction between HSF2BP and BRCA2 might contribute to control the levels of BRCA2 in cells.

## 10. BRCA2 Variants Detected in Patients with Cancers Are Mutated in Folded or Disordered Regions: Which of These Variants Are Pathogenic?

Germline pathogenic variants in the *BRCA2* cancer susceptibility gene result in an increased lifetime risk of breast, ovarian, and other cancers. Tumors forming in patients with a pathogenic variant in *BRCA2* exhibit significant structural and numerical chromosomal defects. Because BRCA2 is directly involved in HR-mediated repair of DSB, inter-strand crosslinks, and replicative DNA lesions, the observed structural chromosomal alterations are thought to derive from the absence of RAD51-mediated BRCA2 DNA repair activity. In contrast, whole-chromosomal defects detected in *BRCA2* mutant tumors and deficient cells are proposed to result from aberrations in both chromosome segregation and cell division. Assessing the pathogenicity of non-truncating missense variants of the *BRCA2* gene, by identifying defects caused by the encoded BRCA2 mutant, is essential to improve the follow-up and treatment of patient families. In particular, tumors with *BRCA2* pathogenic variants associated with defective HR are sensitive to Poly(ADP ribose) polymerase inhibitors, the efficacy of which is mediated through synthetic lethality with BRCA2 loss-of-function in cancer cells [80,81].

Interpretation of variants is currently based on a combination of population, computational, functional and segregation analyses. Each variant is assigned to one of the five following classes: benign, likely benign, uncertain significance, likely pathogenic and pathogenic [82]. First, nonsense or frameshift variants within the coding exons of *BRCA2*, as well as variants in the canonical splice site sequences of *BRCA2*, strongly alter the structure of the protein product and are presumed to confer loss-of-function [83,84,85,86,87]. They are generally classified as pathogenic. However, they may retain (partial) functionality through the expression of alternative protein isoforms, leading to possible incorrect risk estimations [87,88]. Second, missense variants can also be classified as benign or pathogenic when detected in a large set of families. However, the vast majority of these variants are individually rare in both the general population and cancer patients. Figure 7A represents the distribution and significance of missense variants in the French population, including those exhibiting splicing defects, along the *BRCA2* gene [89]. Variants are detected in the whole gene, with no preferred hot spot, and most of these variants are still of uncertain significance. 

Focusing on missense variants in BRCA2 IDRs, only W31S is nowadays being classified as pathogenic in the French variant database based on co-segregation analysis (S. Caputo, personal communication). This variant is localized in exon 3, and impairs BRCA2 binding to PALB2, an essential event in HR ([37]; Figure 7B). In frame deletion of the whole, exon 3 was also classified as pathogenic based on clinical, functional, and co-segregation data [90,91], whereas variants causing only partial production of a *BRCA2* transcript deleted from exon 3 have variable impacts on tumorigenesis [92]. Similarly, in frame deletion of exon 11 impairs HR and was classified as likely pathogenic in ClinVar [87].

We recently showed that S206C and T207A lead to chromosomal instability including aneuploidy as observed in BRCA2 mutated tumors, which we proposed could have an impact on cancer [57]. Ser206 and Thr207 are encoded by exon 7. This exon is not in frame, and S206C induces exon 7 skipping at 46%, thus impairing BRCA2 function because of both the mutation and the low level of the variant expression [93]. However, the community is questioning the different types of variants of exon 7 because they often cause the expression of different transcripts including an in-frame delta 4-7 isoform (BRCA2_delExons4-7_), which is transcribed into an HR competent protein [87,94,95]. However, no clinical data support the classification of the variants S206C, T207A or BRCA2_delExons4-7_ as benign or pathogenic [93,96]. In contrast, in frame deletion of exon 12 that causes a strong decrease in HSF2BP binding does not impact HR efficiency [97,98].

**Figure 7 biomolecules-11-01060-f007:**
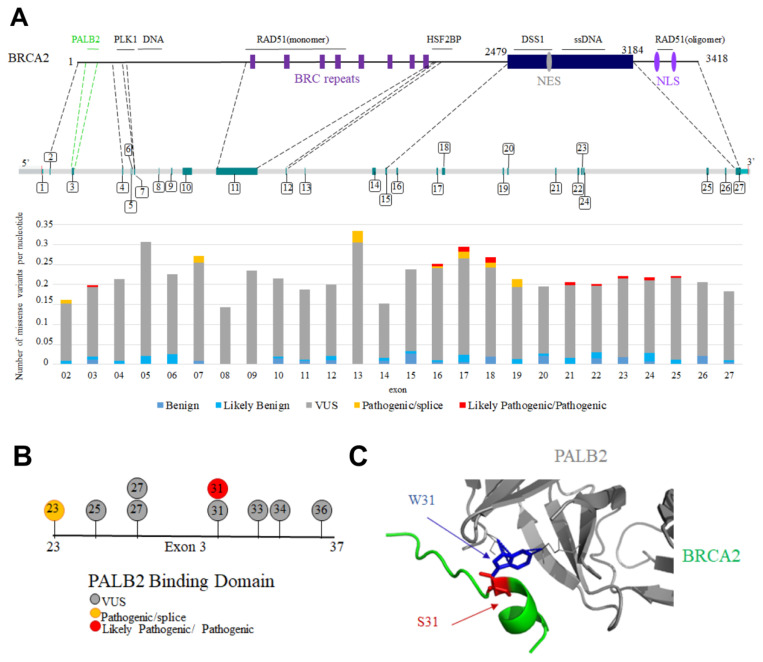
*BRCA2* missense variants located in exons coding for disordered regions are generally of uncertain significance. (**A**) Distribution of the BRCA2 missense variants detected in patients with cancers in the French population. In the upper panel, BRCA2 protein and gene organization are schematized. Binding regions described in this review are indicated above the protein scheme. Protein regions encoded by exons of interest for this review are connected to their corresponding exons through dotted lines. Exon numbers are indicated. In the lower panel, the number of missense variants is represented per nucleotide and per exon of *BRCA2*, as deduced from the analysis of the French database. (**B**) *BRCA2* variants mutated in the PALB2 binding site encoded by exon 3; (**C**) 3D structure of the interface between BRCA2 and PALB2, with a focus on the variant W31S. The wild-type Trp31 (blue) was mutated to a Ser (red) in the 3D structure referenced as PDB 3EU7 using the online server Missense3D [99].

As stated above, most high-throughput functional assays performed to improve the interpretation of variants are based on the evaluation of the HR function of BRCA2 [100,101,102,103]. Using this type of assays, pathogenic variants were identified in the *N*-terminal disordered region binding to PALB2 and in the *C*-terminal folded DBD binding to ssDNA and DSS1. This is consistent with the fact that these BRCA2 regions are essential for HR. However, other functions of BRCA2 were revealed, for example in the protection of stalled replication forks [3], during conflicts between DNA transcription and replication [104], at DNA-RNA hybrids [105,106] or in mitosis [57]. Disordered regions of BRCA2 contribute to these mechanisms, which can then be regulated by post-translational modifications. Defects in these emerging functions might also promote chromosome instability and tumorigenesis [107]. Thus, the concept of BRCAness should nowadays be redefined to include BRCA2 new functions, and the role of variants of IDRs should be reassessed to take into account new mechanisms potentially leading to tumorigenesis.

## 11. Conclusions

Here, we illustrated how BRCA2’s large disordered regions serve to recruit kinases, phosphatases as well as other proteins involved in genome integrity. We described how the *N*-terminal and central regions of BRCA2 bind to PALB2, RAD51 and DNA, thus contributing, together with the *C*-terminal folded DNA binding domain, to the function of BRCA2 in DNA repair by HR. Testing the HR capacity of cancer-associated variants is still the most common approach used to experimentally validate the pathogenicity of these variants, i.e., their impact on tumorigenesis. In BRCA2 IDRs, a variant affecting Trp31 is nowadays being classified as pathogenic and impairs HR. A set of variants characterized by an intermediate HR efficiency (hypomorphic variants) were also identified, which are still not classified in the case of BRCA2, but could correspond to a new class associated with an intermediate cancer risk [108]. Other functional defects of BRCA2 might reveal additional variants with a pathogenic impact in IDRs. Indeed, the disordered regions of BRCA2 are involved in BRCA2 oligomerization, thus potentially indirectly regulating a number of other functions. In particular, mutating the BRC4 repeat was reported as significantly reducing BRCA2 ability to self-associate [23]. Further work is needed in order to identify the intramolecular interactions involved in BRCA2 oligomerization as well as their modes of regulation during DNA repair or throughout the cell cycle. BRCA2 central region is also able to recruit the meiotic protein HSF2BP, whose overexpression triggers BRCA2 degradation. Thus, BRCA2 IDRs might contribute to control BRCA2 level in cells. 

Finally, BRCA2 IDRs are phosphorylated by kinases regulating DNA repair and mitosis. These phosphorylation events are still poorly described. However, they can be functionally critical and required for the coordination of the interphase and mitotic functions of BRCA2. Further studies combining biophysics and cell biology approaches are now needed to identify new phospho-dependent partners for the reported disordered and conserved motifs in BRCA2, as well as to elucidate the function of other less characterized conserved motifs. Such studies will enlarge our knowledge of the molecular events contributing to genome integrity. It will also enlarge the set of experimental assays available to assess the pathogenicity of variants. It will finally stimulate the design of new therapeutic strategies targeting cancers triggered by specific variants.

## Figures and Tables

**Figure 1 biomolecules-11-01060-f001:**
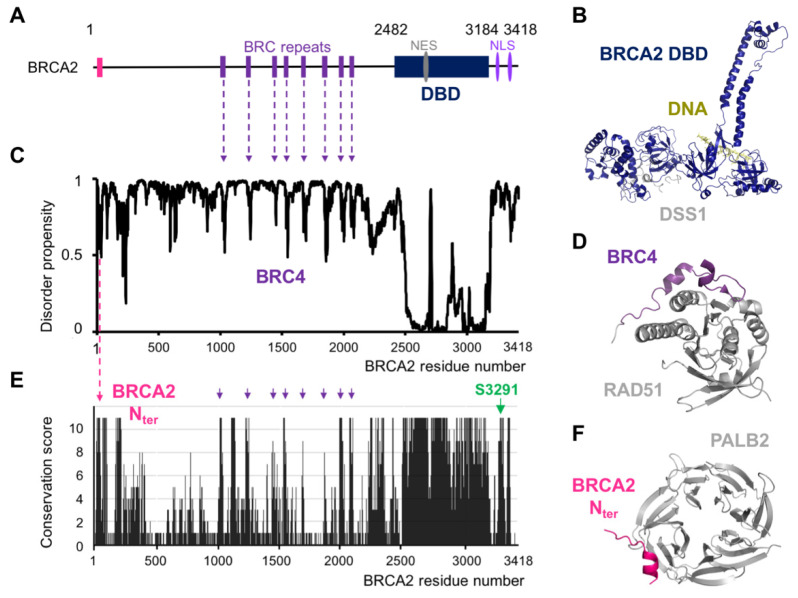
BRCA2 is predicted as mostly disordered, and small conserved motifs fold upon binding. (**A**) General view of BRCA2 structural organization. The folded DNA-binding domain (DBD) is displayed as a dark blue rectangle. The Nuclear Export Signal (NES) and Nuclear Localization Signal (NLS) are marked in grey and purple, respectively. The PALB2 and recombinase binding motifs are represented as pink and violet bars, respectively. (**B**) 3D model of the human DBD (dark blue) bound to an 8nt-ssDNA (limon green) and the small acidic protein DSS1 (grey). This model was built by homology based on the mouse DBD 3D structure, referenced as 1MJE at the PDB; (**C**) disorder propensity as a function of BRCA2 residue number. This disorder score was calculated using the Webserver SPOT-Disorder2 [20]. Scores of 0 and 1 correspond to ordered and disordered residues, respectively. The position of the BRCA2 motif BRC4 is indicated in violet. (**D**) 3D structure of BRC4 (violet) bound to the ATPase domain of RAD51 (grey). This cartoon view corresponds to the 3D structure referenced as 1N0W. (**E**) Conservation of BRCA2 residues calculated from an alignment of BRCA2 sequences from fishes to human. The conservation score was calculated using JALVIEW. Scores of 0 and 11 correspond to non-conserved and strictly conserved residues, respectively. The positions of the PALB2 binding domain (BRCA2_Nter_) and of residue Ser3291 are indicated in pink and green, respectively. (**F**) 3D structure of BRCA2_Nter_ (pink) bound to PALB2 (grey). This cartoon view corresponds to the 3D structure referenced as 3EU7.

**Figure 2 biomolecules-11-01060-f002:**
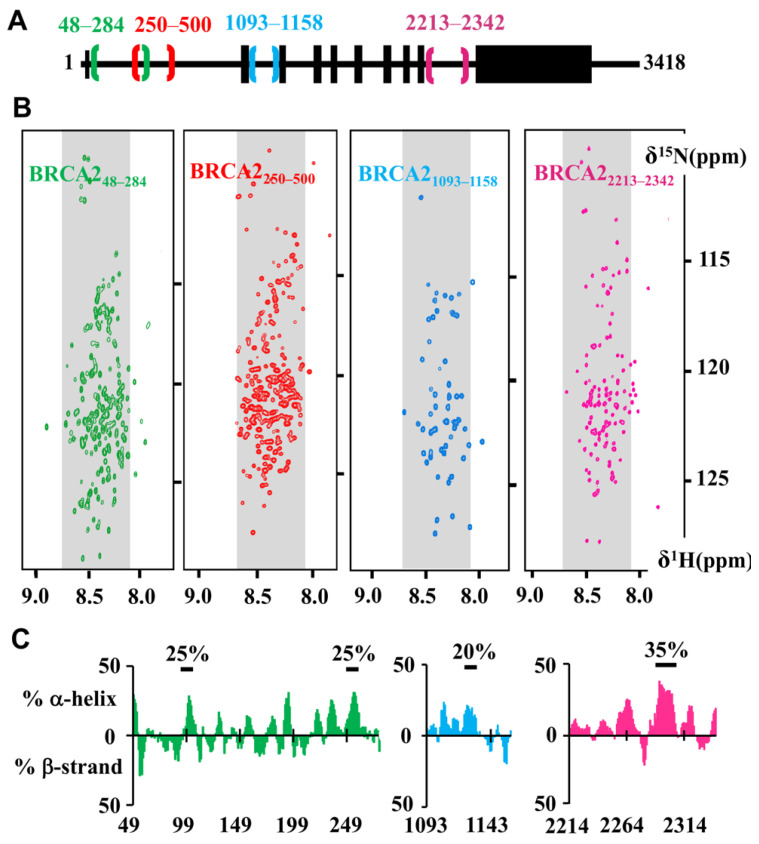
NMR analysis demonstrates that the conserved motifs BRCA2_48–284_, BRCA2_250–500_, BRCA2_1093–1158_, BRCA2_2213–2342_ are disordered in solution. (**A**) Localization of the four fragments in the BRCA2 sequence. (**B**) 2D NMR ^1^H-^15^N Heteronuclear Single Quantum Coherence (HSQC) spectra of the four fragments. These spectra were recorded on ^15^N-labeled BRCA2_48–284_ at 200 μM in HEPES 50 mM, NaCl 75 mM, EDTA 1 mM, DTT 5 mM, pH 7.0 (green, [53]), ^15^N-labeled BRCA2_250–500_ at 200 μM in phosphate 20 mM, NaCl 250 mM, pH 7.0 (red, 600 MHz CEA Saclay), ^15^N-labeled BRCA2_1093–1158_ at 200 μM in 50 mM HEPES, 50 mM NaCl, 1 mM EDTA pH 7.0 (blue, 700 MHz CEA Saclay) and ^15^N-labeled BRCA2_2213–2342_ 500 µM, in HEPES 50 mM, NaCl 50 mM, EDTA 1 mM, pH 7.0 (pink, [54]). Cysteines of BRCA2_48_–_284_ (C132A, C138A, C148A, C161A) and BRCA2_250–500_ (C279A, C311S, C315A, C341A, C393S, C419S, C480S) were mutated in order to favor their solubility. Grey backgrounds define the ^1^H frequencies typical for disordered residues. Localization of most ^1^H-^15^N peaks in the grey regions highlights the disorder propensity of these BRCA2 fragments; (**C**) Secondary structure propensity of three of these fragments. Propensities to form α-helices or β-strands were calculated for BRCA2_48–284_ (green), BRCA2_1093–1158_ (blue) and BRCA2_2213–2342_ (pink) from their ^1^HN, ^15^NH, ^13^CO, ^13^Cα and ^13^Cβ NMR resonance frequencies using the ncSPC server [56]. Bold black lines underline the presence of transient secondary structure with the indicated percentage. BRCA2_48–284_ contains two shorts transient α-helices formed by residues 100 to 110 and 255 to 260, BRCA2_1093–1158_ exhibits one transient α-helix between residues 1122 and 1128 and BRCA2_2213–2342_ shows one transient α-helix between residues 2292 and 2303.

**Figure 3 biomolecules-11-01060-f003:**
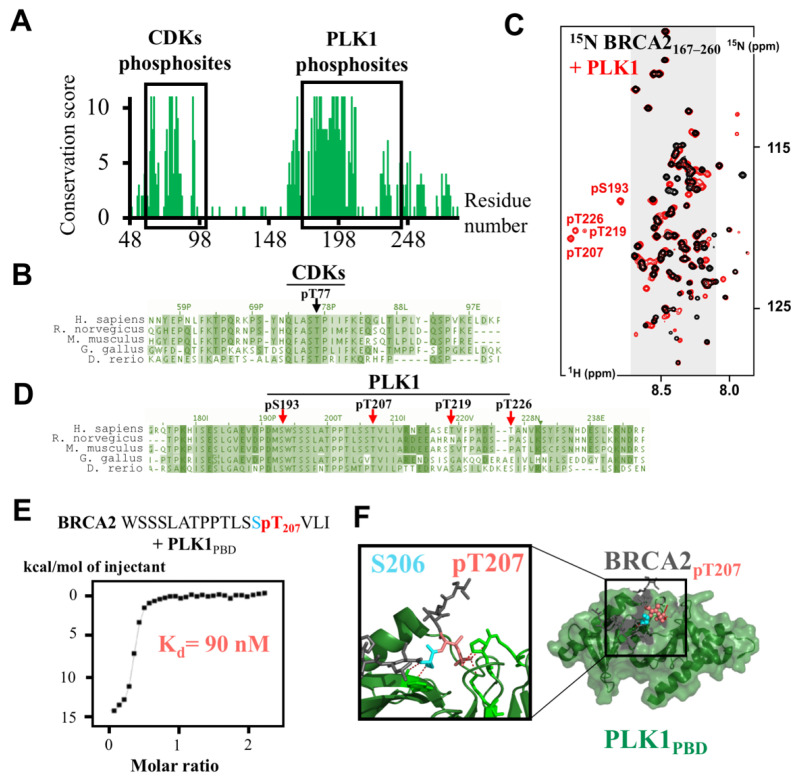
BRCA2_48–284_ contains conserved CDKs and PLK1 phosphorylation sites and two PLK1 docking sites. (**A**) BRCA2_48–284_ conservation profile, calculated as in Figure 1, shows two conserved patches: BRCA2_63-93_ and BRCA2_164–230_. These patches contain predicted as well as experimentally identified CDKs and PLK1 phosphorylation sites, marked by black boxes. (**B**,**D**) Alignments of the human, mouse, rat, chicken and zebra fish BRCA2 sequences corresponding to the regions boxed in (**A**) illustrate the conservation of the reported CDKs and PLK1 phosphorylation sites. Arrows point to the CDKs (black) and PLK1 (red) phosphorylation sites in these regions: pThr77 [7,59], pSer193 [7,59,60], pThr207, pThr219 and pThr226 [57]. (**C**) NMR analysis identified four PLK1-dependent phosphosites: pSer193, pThr207, pThr219 and pThr226 [57]. ^1^H-^15^N SOFAST-HMQC spectra of BRCA2_167–260_ (50 μM) were recorded before (black) and after (red) 12 hrs of in vitro phosphorylation by PLK1 (150 nM). (**E**) Isothermal Titration Calorimetry (ITC) thermogram revealed binding of a BRCA2 peptide containing pThr207 to the PBD domain of PLK1 (PLK1_PBD_; [57]). (**F**) 3D structure of BRCA2_200–209(pThr207)_ bound to PLK1_PBD_ identified critical intermolecular interactions involving BRCA2 Ser206 and pThr207 [57]. In BRCA2_48–284_, two PLK1 docking sites were identified, centered on either pThr77 [60] or pThr207 [57]. The X-ray structure of BRCA2_199–210(pThr207)_ bound to PLK1_PBD_ was recently solved (PDB code: 6GY2). Within BRCA2_199–210(pThr207)_ (grey, blue and salmon), side chains of BRCA2 Ser206 (blue) and pThr207 (salmon) directly bind to the two Polo-box motifs of the PBD (green), and drive the specificity of the interaction. PBD residues in interaction with pThr207 and Ser206 are displayed as green sticks, and red dotted lines mark the hydrogen bonds involved in these interactions.

**Figure 5 biomolecules-11-01060-f005:**
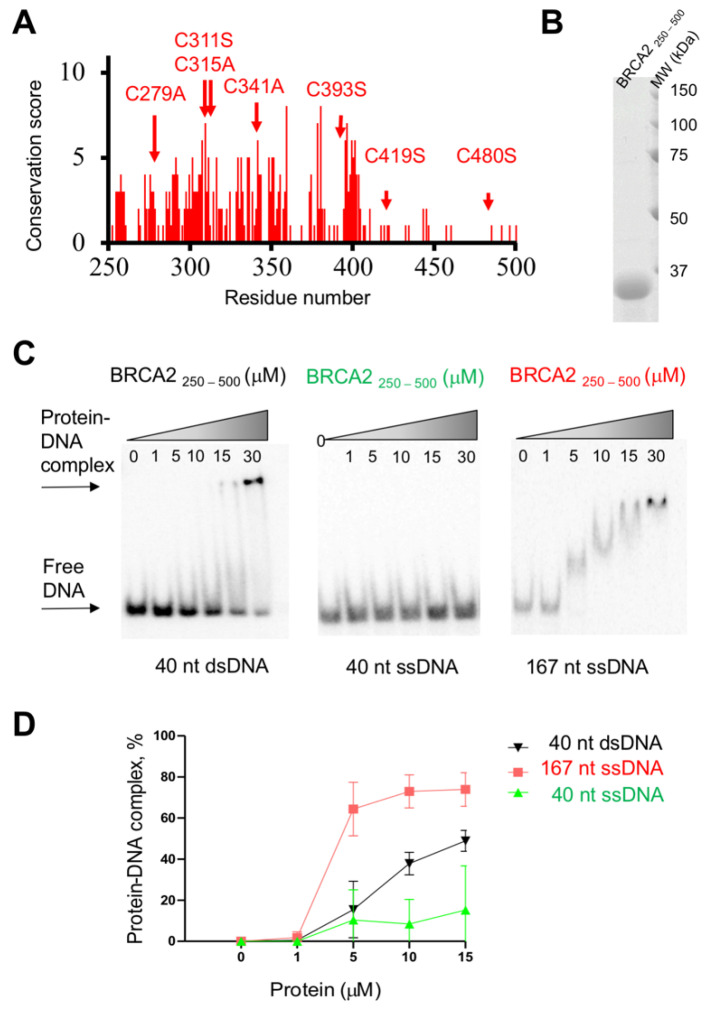
BRCA2_250–500_ contains a DNA binding site. (**A**) Conservation profile of BRCA2_250–500_ obtained as presented in Figure 1. Red arrows indicate the position of the seven cysteines that were mutated into alanine or serine depending on the amino acids found at these positions in the homologous BRCA2 sequences: C279A, C311S, C315A, C341A, C393S, C419S, C480S. (**B**) SDS-PAGE of purified recombinant BRCA2_250–500_. (**C**) BRCA2_250–500_ binds to different DNA substrates, as observed by electrophoretic mobility shift assays. Increasing amounts of BRCA2_250–500_ were incubated at the indicated concentrations with 0.2 µM (nucleotide) ^32^P-labelled DNA substrates for 1 h at 37 °C. The protein–DNA complexes were resolved on 6% native polyacrylamide gels in 1X TAE buffer at 70 V for 75 min. The gels were dried and analyzed with a Typhoon PhosphoImager (Amersham Biosciences) using Image Quant software (GE Healthcare). (**D**) Quantification of the EMSA experiments, revealing that BRCA2_250–500_ binds to both ssDNA and dsDNA. In all gels, the ratio of protein-DNA complexes was calculated as the percentage of bound vs. free DNA. The experiments were repeated three times for each DNA substrate.

**Figure 6 biomolecules-11-01060-f006:**
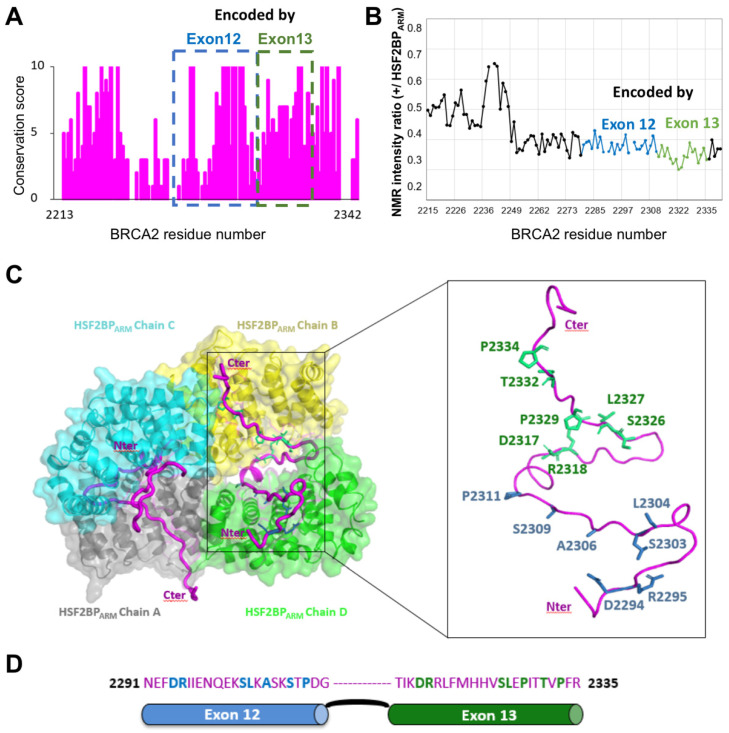
BRCA2_2213–2342_ interacts with HSF2BP, a protein essential for meiotic HR. (**A**) BRCA2_2213–2342_ conservation profile, as calculated in Figure 1. Regions encoded by exons 12 and 13 are boxed in blue and green, respectively. (**B**) NMR analysis of BRCA2_2213–2342_ interaction with the Armadillo domain of HSF2BP (HSF2BP_ARM_) [54]. 2D ^1^H-^15^N HSQC spectra were recorded on ^15^N-labeled BRCA2_2213-2342_, either free or in the presence of HSF2BP_ARM_ (1:1 ratio, knowing that 1 BRCA2 peptide binds to 2 HSF2BP_ARM_), at 950 MHz and 283 K. Ratios of peak intensities in the two conditions are plotted as a function of BRCA2 residue number. The points and curve fragments in blue and green correspond to residues encoded by exons 12 and 13 of BRCA2, respectively. (**C**) Crystal structure of the complex between BRCA2_2291–2342_ and HSF2BP_ARM_, illustrating how the HSF2BP_ARM_ dimers, formed by chains A (grey) and D (green) and chains B (yellow) and C (cyan), are held together through their interactions with the two BRCA2 peptides (magenta) [54]. The HSF2BP_ARM_ domains are represented as both cartoon and surface, whereas the BRCA2 peptides are displayed as tubes. In the zoom view, only one BRCA2 peptide is shown. Its *N*-terminal region interacts with one HSF2BP_ARM_ domain through motif 1 (blue sticks), and its *C*-terminal region interacts with another HSF2BP_ARM_ domain through motif 2 (green sticks). Only the side chains of residues that are conserved in BRCA2 from fishes to mammals and are similar between motifs 1 and 2 are displayed. (**D**) Sequence of BRCA2 motifs 1 and 2, encoded by exons 12 and 13, respectively. Each motif binds to one HSF2BP_ARM_. Residues conserved in BRCA2 from fishes to mammals and similar between motifs 1 and 2 are colored in blue and green, respectively.

## Data Availability

Not applicable.

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
