# Peer review of "Intrinsic Disorder and Phosphorylation in BRCA2 Facilitate Tight Regulation of Multiple Conserved Binding Events"

_biomolecules, 2021, doi:10.3390/biom11071060_

Round 1

Reviewer 1 Report

This review focuses on an interesting subject, the role of intrinsically disordered regions of BRCA. Several examples are described in detail with the aim to link biochemical data with residue-resolved information on the disordered regions of BRCA. The review can thus be of interest for the scientific community and it is expected to stimulate further progress.

A few minor comments for improvement:

The authors use one disorder predictor (SPOT-Disorder). Since protein intrinsic disorder is the main focus of the review it would be interesting to compare the results of this predictor with predictions obtained using at least a few of the other tools available (such as PONDR, IUPRED, etc.). This would allow the authors to perform a critical assessment of disorder predictions by evaluating regions of the protein in which all of them are in agreement from regions in which it is more difficult to make predictions.

The review makes extensive use of acronyms; these should be reduced to a minimum if possible.

Reviewer 2 Report

The review article “Intrinsic disorder and phosphorylation in BRCA2 facilitate
tight regulation of multiple conserved binding events” by Julien et al., highlights the functional importance of BRCA2 in cancer. The review is well-written and covers the various aspects of BRCA2 biology in cancers. Moreover, it is appealing of how the IDRs are playing critical roles with binding to other proteins.

1) There are many results (or figures) that needed to be accompanied with permission statement. They may be from authors own work, but already published work may have some copyright restrictions. Please look into this.

2) No abbreviation of BRCA2/BRC repeats, MPB, RPA etc have been provided.

3) Page 9 (4th paragraph): “The region between BRC1 and BRC2, including BRCA21093-1158, is phosphorylated by PLK1 during mitosis.” This should be BRC21093-1158 rather than BRCA21093-1158

4) There should be a space in between to500 in this sentence “The BRCA2 region from residue 250 to500 also contributes to DNA binding”.
